# Milk, Fertility and Udder Health Performance of Purebred Holstein and Three-Breed Rotational Crossbred Cows within French Farms: Insights on the Benefits of Functional Diversity

**DOI:** 10.3390/ani11123414

**Published:** 2021-11-30

**Authors:** Julien Quénon, Marie-Angélina Magne

**Affiliations:** 1Université de Toulouse, INRAE, UMR 1248 AGIR, 31320 Castanet-Tolosan, France; 2Université de Toulouse, INRAE, INPT, INP-EI Purpan, ENSFEA, UMR 1248 AGIR, 31320 Castanet-Tolosan, France; marie-angelina.magne@inrae.fr

**Keywords:** dairy crossbreeding, performances, trade-offs, functional diversity

## Abstract

**Simple Summary:**

When implementing dairy crossbreeding in purebred Holstein (HO) herds, farmers expect to improve the overall herd performance. However, they lack knowledge about how to manage and benefit from the diversity of genetic classes generated by three-breed rotational crossbreeding, which firstly refers to the cohabitation of purebred HO and first- and second-generation crosses (F_1_ and G_2_, respectively) within the herd. This study aimed to compare milk production, reproduction and udder health performance of HO, F_1_ and G_2_ cows, and to estimate how their combination in different proportions in the herd affects its profitability. We found that HO, F_1_ and G_2_ had different and complementary performance profiles, with two main trends. First, HO had higher milk yield, while F_1_ and G_2_ crosses had better fertility performance. Second, F_1_ had win-win trade-offs between milk production, fertility and udder health compared to HO and G_2_. We showed that HO-F_1_ or HO-F_1_-G_2_ (below 30%) mixed herds could be more profitable than purebred HO or fully crossbred herds with a conventional milk price. These findings can be used for advising purebred HO farmers who wonder about the benefits and the ways of managing the diversity of animal entities generated by the use of dairy crossbreeding in their herds.

**Abstract:**

Using three-breed rotational crossbreeding in a purebred Holstein (HO) herd raises two questions: Do the different genetic classes of cows generated by crossbreeding perform differently? Are there any economic benefits of combining them within a herd? This study aimed at comparing the performance between the different genetic classes resulting from the use of three-breed rotational crossbreeding, and simulating the effect of combining them on herd profitability. Based on a dataset of 14 French commercial dairy herds using three-bred rotational crossbreeding from a HO herd over a 10-year period, we defined three genetic classes according to the theoretical value of heterosis and the percentage of HO genes. We performed linear models and estimated least square means to compare HO cows and the first and second generation of crosses (F_1_ and G_2_, respectively) on eight performance characteristics related to milk yield and solids, udder health and fertility. We used these to simulate profitability of five herd compositions differing according to HO, F_1_ and G_2_ proportions. We showed that HO, F_1_ and G_2_ cows had different and complementary performance profiles. HO had a win-lost trade-off between milk yield and fertility, G_2_ had the opposite trade-off and F_1_ had a win-win trade-off. Differences regarding milk solids and udder health were less clear-cut. We highlighted that combining HO with F_1_ or with both F_1_ and G_2_ (below 30%) could be more profitable than using purebred HO or crossbred herds in a conventional milk price scenario. These findings provide evidence on the benefits of functional diversity generated from the use of dairy crossbreeding in dairy herds.

## 1. Introduction

Using three-breed rotational crossbreeding in dairy herds is an appealing and relatively fast option to improve the functional traits of cows within high-yielding purebred herds and to help dairy farmers transitioning towards low-input dairy cattle systems [1,2,3]. Here, functional traits refer to characters of an animal that increase efficiency by reduced costs of input rather than by higher outputs of products (i.e., milk and meat) [4]. Functional traits that receive particular attention in dairy crossbreeding research are fertility, udder health and feed efficiency [5,6,7]. Despite its potential benefits through both breed complementarity and heterosis, using three-breed rotational crossbreeding still remains quite uncommon in France [8] and in many Western countries [2,9,10]. This situation has often been attributed to a lack of references on crossbreeding’s benefits, which, however, has been gradually filled by research. Indeed, some studies have been estimating additive and non-additive genetic parameters of crossbreeding (i.e., heterosis and recombination loss) for production and functional traits [9,10] and assessing phenotypical cows’ performance for specific three-breed crossbreeding programs [11,12,13]. To a lesser extent, others have also assessed the economic performance of crossbred herds for specific three-breed crossbreeding programs [14,15]. All these studies share a common approach: they focused on one or several specific crossbreeding programs, i.e., on specific combinations of dairy breeds. This built over the years an increasingly precise comparative of benefits and disadvantages of each program compared to purebred parental populations, according to the production systems [2,16], thus providing references toward dairy farmers on the available “genetic offer” for dairy crossbreeding. Finally, the few studies assessing the effect of three breed rotational crossbreeding on herd performance proceeded by simulation modeling and considered that one single crossbreeding program was introduced and managed integrally in the herd [14,15]. However, many dairy cattle farmers introduce and manage crossbreeding in a customized way and following a trial-and-error proceeding, which lead them to manage—temporarily or lastingly—several crossbreeding programs simultaneously [17], while keeping part of their herd purebred [14]. Consequently, their herd can be composed of diverse purebred and crossbred genotypes: introducing three-breed crossbreeding in a Holstein herd led to manage genetically diverse animal classes, regardless of crossbreeding programs. Hence, critical questions for dairy cattle farmers who manage the transition of a herd from pure-breeding to crossbreeding are: in which way the different genetic classes of cows within the herd differ in their performance? Do crossbred cows perform better than purebred cows in the herd? Do crosses of n + 1 generations maintain an advantage over those of n generations? Is it beneficial to herd profitability to combine these genetic classes? Answering these questions would contribute to address the issue of “functional diversity”, defined as “the variety of functions played by individual production entities involved in overall performance” [14] and that may play a more critical role in overall performance than genetic or inter-specific diversity as such [14,15,16].

Therefore, the aim of the study was two-fold: (1) first, to assess and compare the performance for milk traits and fertility and udder health traits (referred to below as functional traits) between the different genetic animal classes resulting from the use of three-breed rotational crossbreeding, from a sample of French commercial dairy cattle farms practicing it over a 10-year period; (2) second, to simulate how different combinations of these genetic animal classes affect herd profitability.

## 2. Materials and Methods

### 2.1. Farm Sampling

The study draws on an original sample of 26 commercial dairy farms in France sampled as they managed three-breed crossbreeding and had at least 33% crossbred cows (all generations combined) and at least second-generation lactating cows in 2018 [17]. From these farms, we selected 14 in which herds were initially composed of purebred Holstein cows (n = 10) or with several pure breeds and prevailing Holstein breed (n = 4). All 14 herds had been enrolled in the French Milk Record Organization from 2008–2018. Data available included milk, calving, fertility and udder health performance, as well as the breed and genealogy of females. These 14 dairy farms (8 organic and 6 conventional) were located in four main regions of dairy production in mainland France: Brittany (n = 4), Auvergne-Rhône-Alpes (n = 4), Hauts-de-France (n = 3) and Normandy (n = 3). The farms varied in size, forage systems and milk yield in 2008 and 2018 (Table 1).

### 2.2. Definition of Genetic Classes

Breed combinations varied greatly among the 14 herds (Figure 1), and more importantly, within each herd (Figure A1). Due to the sampling criteria, purebred Holstein was the most common genotype among the sampled herds. Therefore, we used two indicators to define the genetic classes that resulted from using three-breed crossbreeding. The first one was the percentage of Holstein genes (HOg) of each female, which we calculated from the breed and genealogical data [7]. The second indicator was the coefficient of heterosis (H), which we calculated as follows [18]:(1)$$H=1-\sum s_{i}d_{i}$$
where $s_{i}$ and $d_{i}$ are the proportions of sire genes and dam genes, respectively, from breed *i*.

We then removed lactations for genetic classes that were insufficiently distributed among the sampled herds (i.e., third-, fourth- and fifth-generation crosses), which left only three genetic classes: purebred Holstein (HO), first-generation crosses (F_1_) and second-generation crosses (G_2_), which have 100%, 50% and 25% of Holstein genes, respectively (Table 2). In doing so, however, the coefficient of heterosis became non-discriminatory in our database, since it was 100% for both F_1_ and G_2_.

### 2.3. Data Editing

The French Milk Record Organization provided data for lactations of females of sampled herds. Available data included total and 305-day yields, somatic cell count (SCC), parity, lactation length, birth date, calving date, age at calving and length of the dry period. We calculated for each lactation from each cow milk yield (MY, in kg/cow) by either extrapolating or correcting for duration values of total and 305-day milk yield [19]. We then calculated corrected values for fat (FY, in kg/cow) and protein (PY, in kg/cow) yields, as well as fat (FC, in g/kg/cow) and protein (PC, in g/kg/cow) contents. We used three categories to express values for duration of the dry period of (n − 1) lactation: <50 days, 50–70 days and >70 days [20]. First, we calculated mean SCC for lactations with at least six SCC values available, then somatic cell score (SCS) as $log_{2}\left(\frac{mean\;\mathrm{SCC}}{100,000}\right)+3$. We calculated calving to first service interval (CFS, in days) as the number of days from the start of lactation to the first service, and calving interval (CI, in days) as the number of days from the start of (n − 1) lactation to the start of lactation n.

Since SCS and reproduction data were missing for some lactations (41% and 63%, respectively), for which production data were available, we created three separate datasets (Table 3). Dataset 1 consisted of milk performance data (i.e., MY, FY, PY, FC and PC) for 6672 lactations from 2730 cows. Dataset 2 consisted of SCS data for 3947 lactations from 2074 cows. Dataset 3 consisted of fertility performance data (i.e., CFS, CI) for 2449 lactations from 1411 cows. The breeds’ combinations that were the most represented in datasets 1, 2 and 3 among F_1_ and G_2_ classes were the following: HO × Montbéliarde (MO), HO × Viking Red (VR), HO × Simmental and HO × Brown Swiss (BS) for F_1_, which represented about 54%, 9%, 8% and 6% of total sampled lactations in dataset 1, respectively, and (HO × MO) × BS, (HO × MO) × SR and (HO × SR) × MO for G_2_, which represented about 33%, 29% and 5% of total sampled lactations, respectively. The contributions to the three datasets of the diverse breed combinations involved in F_1_ and G_2_ crosses are detailed in Table A1.

### 2.4. Statistical Analyses

We compared the three genetic classes on eight performance traits: five related to milk production (MY, FY, PY, FC and PC), one related to udder health (SCS) and two related to fertility (CFS and CI). We used linear models and estimated least square means (LSM) [21] for genetic classes (i.e., HO, F_1_ and G_2_) within parity (i.e., primiparous and multiparous). We then tested for differences in performance between pairs of genetic classes within parity using Tukey’s tests (*p* < 0.05). We performed all statistical analyses using the emmeans package [22] in RStudio software (v. 4.0.4, RStudio Inc., Boston, MA, USA).

To estimate LSM of performance traits in dataset 1 (milk performance data), we used an adjusted version of the French genetic evaluation model for production traits [23]:(2)$$Y_{ijklmn}=\mu +P_{i}\times G_{j}+H_{k}\times Y_{l}+P_{i}\times M_{m}+P_{i}\times D_{n}+A_{o}+\varepsilon_{ijklmn}$$
where $Y_{ijklmn}$ = observation for the dependent variable, μ = overall population mean,$\;P_{i}\times G_{j}$ = fixed effect of the *j*-th genetic class (*j* = HO, F_1_, G_2_) within the *i*-th parity (*i* = primiparous, multiparous), $H_{k}\times Y_{l}\;$= fixed effect of herd-year for the *k*-th herd (*k* = 1–14) and *l*-th year (*l* = 2009–2018), $P_{i}\times M_{m}$ = fixed effect of the *m*-th calving month (*m* = January to December) nested within parity, $P_{i}\times D_{n}$ = fixed effect of length of the *n*-th previous dry period (n =< 50 days, 50–70 days, >70 days) within parity, $P_{i}\times A_{o}$ = fixed effect of *o*-th value for age at calving within the *i*-th parity (*i* = primiparous, multiparous), and $\varepsilon_{ijklmn}$ = residual error term.

To estimate LSM of performance traits in dataset 2 (somatic cells score data), we used an adjusted version of the French genetic evaluation model for cell count [23]:(3)$$Y_{ijklmn}^{\prime }=\mu^{\prime }+P_{i}\times G_{j}+H_{k}\times Y_{l}+P_{i}\times M_{m}+P_{i}\times D_{n}+P_{i}\times A_{o}+\varepsilon_{ijklmno}^{\prime }$$
where $Y_{ijklmno}^{\prime }$ = observation for the dependent variable, $\mu^{\prime }$ = overall population mean, and $\varepsilon_{ijklmno}^{\prime }$ = residual error term.

Finally, to estimate LSM of performance traits in dataset 3 (reproduction performance data), we used an adjusted version of the French genetic evaluation model for reproduction traits [23]:(4)$$Y_{ijklmo}^{\prime \prime }=\mu^{\prime \prime }+C_{i}\times G_{j}+H_{k}\times Y_{l}+Y_{l}\times M_{m}+A_{o}+C_{i}+\varepsilon_{ijklmo}^{\prime \prime }$$
where $Y_{ijklmo}^{\prime \prime }$ = observation for the dependent variable, $\mu^{\prime \prime }$ = overall population mean,$\;C_{i}$ = fixed effect of *i*-th calving rank prior to insemination (*i* = primiparous, multiparous), and $\varepsilon_{ijklmo}^{\prime \prime }$ = residual error term.

The response variable used for the linear models of the reproduction performance (i.e., CFS and CI) were log-transformed. LSM were then transformed back.

### 2.5. Design of Simulated Herd Compositions and Estimation of Profitability

We considered a herd of 100 cows with several compositions of the three different genetic classes (i.e., combinations of HO, F_1_ and G_2_), based on the demographic dynamics of a crossbreeding herd [14]. Hence, we simulated five herd compositions (Table 4): one purebred Holstein herd (100HO) and four mixed herds including one HO-F_1_ herd at 50% each (50HO-50F_1_) and three mixed HO-F_1_-G_2_ herds (32HO-63F_1_-5G_2_, 18HO-67F_1_-15G_2_ and 10HO-60F_1_-30G_2_).

We calculated the performance of each herd composition for milk yield, fat and protein contents and calving interval as the mean value of LSM’s weighted by the proportions of HO, F_1_, and G_2_ in each herd composition simulated, as follows:(5)$$\bar{x}=\frac{x_{HO}\times w_{HO}+x_{F_{1}}\times w_{F_{1}}+x_{G_{2}}\times w_{G_{2}}}{w_{HO}+w_{F_{1}}+w_{G_{2}}}$$
where $\bar{x}$ is the weighted mean value for a given performance of the herd and $x_{i}$ is the LSM value estimated for the given performance of the *i*-th genetic class (*i* = HO, F_1_, G_2_*)* and is the proportion of cows of the *i*-th genetic class in the herd composition.

Then, for each herd composition simulated, we estimated the income generated by the volume of milk from the cows [24], the economic gains and costs related to the protein and fat contents, the reproduction costs of the cows [25] and finally the economic profitability generated by the milk production taking into account these bonuses/penalties. We considered a conventional milk price scenario in France.

## 3. Results

### 3.1. Performance of Primiparous Cows

Primiparous cows of the three genetic classes had different performance profiles, with “win-win” trade-offs between milk production and fertility traits trade-offs for F_1_ cows compared to HO and G_2_ cows, for which trade-offs were to the detriment of fertility and milk yield, respectively (Table 5). F_1_ had significantly higher FY than both HO and G_2_ (310 vs. 296 and 294 kg, respectively). They also had higher MY (7655 vs. 7050 kg) and PY (245 vs. 240 kg) than G_2_, while not significantly differing from HO. G_2_ primiparous cows had the highest performance for fat and protein contents of the three genetic classes: FC and PC were significantly higher for G_2_ than for F_1_ (+1.4 g/kg for FC and +1.0 g/kg for PC) and significantly higher for F_1_ than for HO (+2.2 g/kg for FC and +0.8 g/kg for PC).

Regarding functional traits, both primiparous F_1_ and G_2_ crosses were more fertile than HO. Both CFS and CI were shorter for F_1_ (+12 and +23 days, respectively) and G_2_ (+17 and +26 days, respectively) than HO, while they did not differ significantly between F_1_ and G_2_. Conversely, there was no significant difference between the three genetic classes for SCS.

### 3.2. Performance of Multiparous Cows

As for primiparous cows, multiparous cows of the three genetic classes had different performance profiles, with F_1_ maintaining a “win-win” trade-off—yet not as favorable as for multiparous cows—between milk production and functional traits compared to HO and G_2_ cows (Table 6). Multiparous F_1_ produced significantly less MY than HO (7596 vs. 7790 kg), but significantly more than G_2_ (7596 vs. 6953 kg). They had higher FY than HO and G_2_ (+14 and +16 kg, respectively) and higher PY than G_2_. Conversely, PY did not differ significantly neither between HO and F_1_, nor between HO and G_2_. FC and PC were significantly higher for F_1_ than for HO (+1.9 g/kg for FC and +0.9 g/kg for PC) and significantly higher for G_2_ than for F_1_ (+0.8 g/kg for FC and +0.4 g/kg for PC).

As for functional traits, CFS did not differ significantly between F_1_ and G_2_. Conversely, CFS was shorter for F_1_ and G_2_ than for HO (−12 days and −13 days, respectively). F_1_ also had shorter calving interval than HO (−18 days), while there was no significant difference neither between F_1_ and G_2_ nor between HO and G_2_. Finally, multiparous F_1_ had significantly lower SCS than HO and G_2_, while the latter did not significantly differ for SCS.

### 3.3. Economic Performance of Herd Compositions Simulated

Mixed herd compositions generated higher profit per dairy cow than the 100% HO herd, with the exception of the mixed herd with a 30% G_2_ share. Economic profitability was +29 €/cow for both the 50HO-50F_1_ and 32HO-63F_1_-5G_2_ mixed herds compared to purebred HO herd (Figure 2), while this was lower, equal to +17 €/cow, for the 32HO-63F_1_-5G_2_ mixed herd (yellow bar in Figure 2).

Conversely, the mixed herd with the largest share of G_2_ (10HO-60F_1_-30G_2_) had a total net economic loss compared to purebred HO herd (−9 €/cow in Figure 2). Although it had the highest economic bonuses related to the increase of fat and protein contents and the improvement of the fertility of the cows (+39, +36 and +41 €/cow, respectively), it did not compensate for the penalty related to the lower milk yield of the crossbred cows compared to the purebred HO ones (−125 €/cow) in a conventional milk price scenario.

## 4. Discussion

### 4.1. Performance Profiles of Purebred HO, F_1_ and G_2_ Crossbred Cows

To discuss the different performance profiles between purebred HO and F_1_ and G_2_ crossbred cows, we based on studies that had focused on breeds’ combinations that were the most represented in datasets 1, 2 and 3 among F_1_ and G_2_ classes (Table A1): HO × MO, HO × VR, HO × Simmental and HO × BS for F_1_ and (HO × MO) × BS, (HO × MO) × SR and (HO × SR) × MO for G_2_.

Regarding the milk performance of F_1_ crosses compared to purebred HO in their first parity, our results are in accordance with those of many studies that reported HO × MO [26,27], HO × BS [28] and HO × SR [2,29,30] had similar milk yields and higher fat and protein yields compared to purebred HO in their first parity. Moreover, our results match those of many studies and conclude that F_1_ have higher fat and protein contents compared to purebred HO in their first parity [29,30]. Conversely, they contrast with studies that also reported lower milk yield [29,31], lower fat and lower protein contents [29,30] for primiparous F_1_ compared to purebred HO ones. Regarding the functional performance, our results contrast with studies [29,31,32] who found lower SCS for F_1_ compared to purebred HO primiparous. Finally, regarding the fertility performance, our results are in accordance with those of many studies [2,29,30,33] that reported shorter DO for F_1_ primiparous compared to purebred HO.

As for multiparous cows, our results are in accordance with most studies that reported that F_1_ crosses had lower milk yield [27,28,29,33], higher fat [28,33] and protein yields [27,28] and higher fat and protein content [29] compared to HO. However, they contrast with fewer studies who reported lower fat and protein yields for multiparous F_1_ compared to purebred HO [2,29]. Regarding SCS, our results contrast with those of studies that reported higher SCS for F_1_ crosses compared to purebred HO [29,33]. Conversely, our results on the fertility performance match those of many studies that reported shorter days open for multiparous F_1_ compared to purebred HO [2,29,32]. However, CFS and CI values for both primiparous and multiparous purebred HO were shorter in our results than the mean CFS and CI were in metropolitan France for purebred HO [25]. In fact, purebred HO in our sample had shorter CFS and CI than 50% of all French dairy farms: 85 vs. 87 days long for CFS and 402 vs. 403 days long for CI.

To our knowledge, very few studies compared G_2_ with F_1_ for the performance characteristics we investigated [12,32,34,35], whether in their first parity or the next ones: they more commonly compare G_2_ with purebred HO [11,27]. Our results report lower milk, fat and protein yields for G_2_ crosses compared to purebred HO in their first parity, and shorter DO, which is in accordance with those studies. Similarly, our results on multiparous cows report lower milk, fat and protein yields for G_2_ crosses compared to purebred HO and shorter DO, which is in accordance with the same studies available [11,27].

Therefore, as most commonly reported in studies on dairy crossbreeding, our findings highlight that HO cows have a “win-lost” trade-off between milk yield and fertility, while G_2_ cows have the opposite trade-off, and F_1_ cows a “win-win” trade-off. By contrast, the advantages of both F_1_ and G_2_ crosses for milk solids (FY and PY) and SCS compared to purebred HO cows were less clear in our results, as they are in the literature on dairy crossbreeding: although F_1_ crosses—and especially multiparous ones—had higher milk solids yields than HO cows, G_2_ crosses do not really maintain such an advantage. This may be because these performance characteristics depend more on feeding and rearing conditions, and therefore, on the systems (commercial farms and experiments) in which the performance characteristics are evaluated.

The performances gaps that we observed between genetic classes are generally smaller than those reported in research on dairy crossbreeding. Comparing multiparous HO and F_1_ cows, we observed a milk yield gap of −196 kg/cow, while other studies reported differences ranging from −347 to −1487 kg/cow [30,31,33]. Similarly, milk yield gap between multiparous HO and G_2_ for MY was −837 kg/cow in our study, while it ranged from −850 to −1466 kg/cow in other studies [27]. The smaller gaps in milk yield and fertility performance that we observed between HO cows and the first- and second-generation crossbreds may be explained by the change in replacement and reform management by dairy farmers during their introduction of dairy crossbreeding in their herd. As the transition process proceeds from a purebred HO herd towards a partially crossbred herd, dairy farmers may reform HO cows with high milk yield but a low fertility performance [17]; consequently, our sample may be constituted accordingly by HO cows with a relatively higher fertility performance and lower milk yield than reported by most of the studies.

### 4.2. What Are the Benefits of Combining HO, F_1_ and G_2_ at the Herd Level?

Research on the benefits of using animal diversity to enhance the performance of livestock systems defines functional diversity as “the variety of functions played by individual production entities involved in overall performance” [36]. Many studies, thus, highlight that functional diversity plays a more critical role in overall performance than genetic or interspecific diversity [36,37,38]. Here, we showed that the performance profiles of the three genetic classes are different. Consequently, when considering the transition of one herd initially composed of purebred HO to rotational crossbreeding, its performance will change at the expense of milk yield and in favor of functional traits. Therefore, the challenge is to identify if combining these genetic classes is beneficial for the overall performance of the herd and which herd composition (i.e., the combination of HO, F_1_ and G_2_) allows to take advantage of the potential complementarity of their performance profiles.

Research on the benefits of using animal diversity to enhance the performance of livestock systems defines functional diversity as “the variety of functions played by individual production entities involved in overall performance” [36]. Our results provide empirical evidence of how such functional diversity is shaped and expressed within a herd and how it interacts with intra-specific diversity, which is still rare in animal production science [36,39]. Here, we showed that the performance profiles of the three genetic classes are different, which means that they are functionally diverse animal entities within the herd.

We also highlighted in our simulations that it may be beneficial to combine such functionally diverse entities. Thus, mixed herds HO-F_1_ or HO-F_1_-G_2_ with 15% G_2_ or less would allow to benefit from complementary performance profiles of HO, F_1_ and G_2_ genetic classes: they combine proportions of the genetic classes in such a way that it both maintains the total milk yield in the herd and increases fat and protein contents, while reducing the operational costs related to cows’ reproduction. Such a search for complementarity between milk yield, content and functional characteristics of genotypes within a herd has already been empirically demonstrated in the case of multi-breed herds in France [37].

Although consistent with previous studies that reported on the economic benefits of dairy crossbreeding in both organic and conventional farming systems [15] or regardless of the intensity level of the farming system [3], our first estimates are more nuanced. Two main reasons can explain that. Firstly, we considered milk price paid in conventional dairy farming in our simulations. However, a higher base milk price, as is the case in organic farming, could offset lower milk yield of G_2_ crossbred cows in terms of the economic value of the total milk production of the herd. Moreover, this may explain why many dairy farmers who introduce dairy crossbreeding in their herds practice or convert to organic farming [1,7,19]. Secondly, we did not integrate the reduction in operational costs generated by the higher concentrate efficiency of crossbred cows [11] in our simulations, nor the added value generated by the valuation of meat products for specific crossbreeding programs [40].

Nevertheless, we provide initial insights about the compositions of crossbred or mixed herds that may be best to target while using rotational crossbreeding. They highlight that mixed HO-F_1_ or HO-F_1_-G_2_ herds with less than 30% of G_2_ cows may be economically more beneficial than purebred HO herds or full-crossbred herds, which are yet often considered as the necessary target for dairy farmers using dairy crossbreeding, and consequently, that are the only herd compositions considered in studies assessing the economic benefits of dairy crossbreeding at the herd level [14,15]. To manage such mixed HO-F_1_ or HO-F_1_-G_2_ herds, dairy farmers can cross dairy breed females with both dairy and beef breed sires, in order to both increase the slaughter value of crossbred male calves [40] and regulate herd replacement for maintaining its mixed composition. Sexed semen can also be used in such mixed herds to have more flexibility in the replacement of the different genotypes [15].

In any case, mixing different genotypes within a dairy herd depends on what the farmer can accept losing and wants to gain. Indeed, based on our findings, three-breed rotational crossbreeding leads to a permanent loss of milk yield and a gain in the reproduction performance at herd level, as reported by most studies [9,14,29]. However, these gain and loss values may vary depending on the breeds used in crossbreeding programs [9,16,17], which cannot be taken into account with our method. Moreover, depending on the strategy dairy farmers adopt to increase the economic value of milk production, they can manage the herd composition differently: they may keep a small percentage of HO cows in the herd to complement loss of milk of F_1_ crosses while benefiting from F_1_ higher fat and protein contents. Alternatively, if economic bonuses for protein and fat contents or more generally base milk price are high enough, dairy farmers may mix F_1_ and G_2_ crosses and move towards a fully crossbred herd. However, this is not the only relevant strategy: other marketing strategies, such as Protected Designations of Origin (PDO) label for milk products, may encourage farmers to substitute HO cows with local pure breeds specified by PDO rules, and therefore, move towards a purebred herd rather than a crossbred one [41]. These results emphasize that an adequate herd composition (i.e., a combination of genetic classes with different trade-off profiles that meet dairy farmers’ objectives), thus, depends on farmers’ personal and business views of the performance of the herd [37,39].

### 4.3. Methodological Comments

In research that addresses the benefits of intra-specific animal diversity in dairy herds, one critical question is to develop a method to classify the different animal entities [36,37,38]. Here, we built genetic classes of cows based on theoretical values for heterosis and percentage of the initial pure breed to assess the effect of crossbreeding on cows’ performance, both crossbred ones and purebred Holsteins that remain in the herd during the transition process to a crossbred herd. Our method of classification of cows’ genotypes within crossbred herds is, therefore, similar to that developed by other studies [7,42]. However, we extended it to second-generation crosses of three-breed rotational crossbreeding, while previous studies only considered first-generation crosses and reciprocal crosses, i.e., two-breed crossbreeding. Our method is promising, as it allows to integrate dairy farmers’ real ways of managing dairy crossbreeding, which are often less linear than introducing and managing one single crossbreeding scheme [17]. Such a method is still rare among studies that investigate dairy crossbreeding and its performance on both experimental [11,27] and commercial farms [2,29]. This raises questions about the applicability of their results under real farm conditions, where the strict application of a crossbreeding program may be limited by on-farm constraints but also the sociotechnical environment of the farms [1].

However, while our method enabled to provide results that are concordant with most of the studies focused on specific crossbreeding programs, such an approach of genetic categorization needs to be consolidated by testing it on a larger farm sample in France and beyond France to assess how robust our results are. We were not able to test the isolated effect of theoretical value for heterosis, as we did not have enough crossbred cows beyond second generation of three-breed rotational crossbreeding in our sample (e.g., G_3_ and further). The heterosis value is yet another deciding factor of crossbreeding performance [18]. Moreover, the imbalance in lactation numbers between genetic classes (i.e., number of HO lactations > F_1_ > G_2_) makes it difficult to interpret the observed differences in standard error values between genetic classes for the estimated performance. It is, thus, difficult to know whether the higher values for G_2_ are a statistical artefact or reflect a greater variability of performance for this genetic class. Most optimal would have been to follow an approach using samples of equivalent size for all crosses; unfortunately, this was practically unreachable and it would not have enabled us to integrate the real management practices of dairy farmers and the performance of cows while transitioning toward rotational crossbreeding. A larger sample should allow integrating more genetic classes, especially crosses beyond the second generation of rotational crossbreeding to test how relevant such genetic categorization is, and to provide a more comprehensive view of the performance of three-breed rotational crossbreeding in the long run. Moreover, a little more homogeneity in crossbreeding programs across sampled farms should allow to integrate the effects of complementarity between breeds into the method of categorizing genetic classes, which is currently lacking. For example, in F_1_ crosses, 50% of genes comes from another breed than HO, but this other breed is not specified in our method. However, depending on whether it is Swiss Brown or Jersey, the characteristics provided by the breed may differ [37], and therefore, the performance of resulting cross with HO may differ too [28,43]. This may explain why we did not observe significant differences between F_1_ and G_2_ or between F_1_ and HO for some traits. This may have been even more difficult if we had investigated performance characteristics, such as slaughter value for calves and cows, for which there is a particularly high variability between breeds [1]. A better consideration of the complementarity effect between breeds in the categorization method is, therefore, a challenge for future investigations.

In our study, we only investigated performance traits related to milk production, fertility and udder health, which is limited to accurately estimate the benefits of dairy crossbreeding on farms [15]. Therefore, there is a need to consider other production performance characteristics, such as those regarding meat production from dairy crossbred females [40] (since crossbred animals have “no real genetic value” [1]) and functional performance characteristics, such as feed conversion efficiency [6], general health [44], longevity or survival [44,45,46]. Finally, our economic estimates were rudimentary and need to be strengthened by integrating more items of production costs (e.g., feeding cost, veterinary and health costs) and by better modeling herd dynamics (e.g., by integrating stochastic herd management events that induce individual variability beyond genetic class [14,15]).

## 5. Conclusions

In this study, we developed a new method for classifying the different animal genetic entities generated by the use of three-breed rotational crossbreeding into purebred HO herds, which enables us to go beyond of the simple crossbreeding program. Based on the theoretical value of heterosis and the percentage of the initial HO purebred, this method was performed on a dataset of 14 French commercial dairy herds practicing three-breed rotational crossbreeding over a 10-year period. We showed that purebred HO and the first and second generation of crossbred cows had different and complementary performance profiles regarding milk production, reproduction and udder health. Thus, HO cows had a win-lost trade-off between milk yield and fertility, while G_2_ cows had the opposite trade-off and F_1_ cows had a win-win trade-off. Regarding milk solids and SCS, the advantages of F_1_ and G_2_ crosses compared to purebred HO were less clear-cut: however, multiparous F_1_ crosses had a better performance than G_2_ crosses. Based on these findings, we provided, for the first time, a few insights on the benefits of combining these functionally diverse animal genotypes at herd level. In a conventional milk price scenario, we highlighted that mixed HO-F_1_ and HO-F_1_-G_2_ herds with lesser that 30% of G_2_ may be economically more beneficial than purebred HO herds than fully three-breed rotational crossbred herds, which are yet often considered as the necessary target for dairy farmers using dairy crossbreeding. However, future studies with greater size sample should verify these results and consolidate the method of classification of animal genetic entities developed here.

## Figures and Tables

**Figure 1 animals-11-03414-f001:**
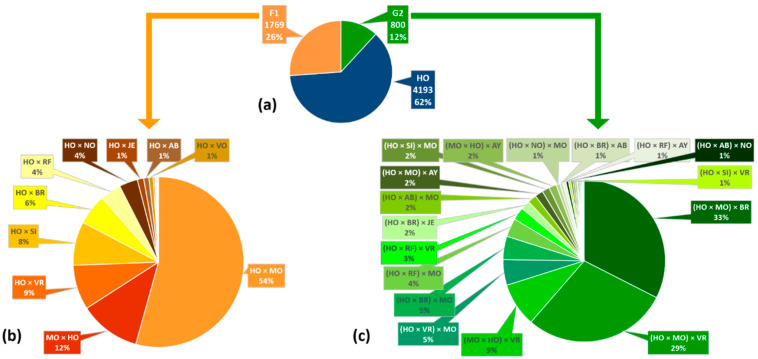
Different breed combinations for lactations data in dataset 1. (**a**) Number of lactations from the three genetic classes, i.e., HO, F_1_ and G_2_. (**b**) Breed combinations for lactations in dataset 1 from first-generation crosses (n = 1769). (**c**) Breed combinations for lactations in dataset 1 from second-generation crosses (n = 800). HO = purebred Holstein; F_1_ = first-generation crosses (50% HO); G_2_ = second-generation crosses (25% HO). AB = Abondance; AY = Ayrshire; BR = Brown Swiss; JE = Jersey; MO = Montbéliarde; NO = Normande; RF = Belgian Red; SI = Simmental; VO = Vosgienne; VR = Viking Red.

**Figure 2 animals-11-03414-f002:**
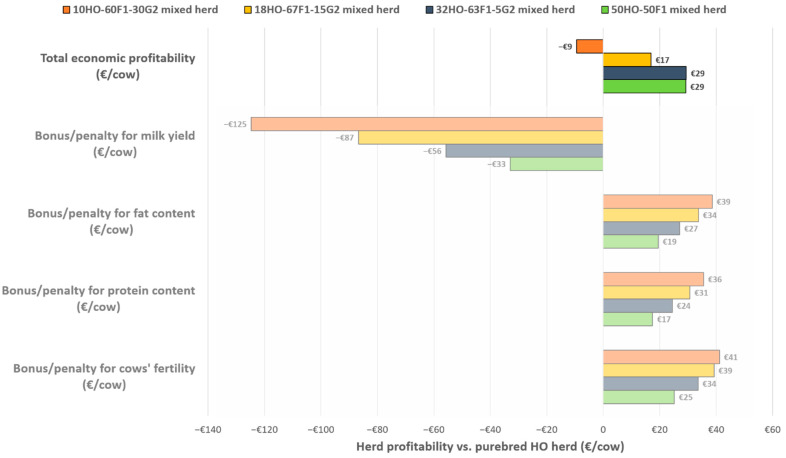
Differences in estimated profitability of four mixed herd compositions compared to the 100% Holstein herd in Euros/cow or Euros/1000 kg of milk. Total economic profitability for each herd was calculated as the sum of economic bonuses/penalties for milk yield, fat and protein contents and cows’ fertility.

**Table 1 animals-11-03414-t001:** Main characteristics (median, minimum and maximum values) of the 14 sampled dairy cattle farms in 2008 and 2018.

Characteristic	2008			2018		
	Med	Min	Max	Med	Min	Max
Farm size (ha)	80	36	330	98	43	225
Grassland area (% of LFA ^1^)	88	23	100	91	16	100
Silage maize area (% of LFA ^1^)	12	0	77	9	0	84
Herd size	44	25	90	70	25	139
Farm milk yield (kg/cow/year)	7363	6303	9694	5997	4716	8513

^1^ LFA = Livestock Feeding Area.

**Table 2 animals-11-03414-t002:** Characteristics of genetic classes as a function of the combination of the percentage of Holstein genes (HOg) and coefficient of heterosis (H).

Genetic class ^1^	HO	F_1_	G_2_	G_3_	G_4_	G_5_
HOg (%)	100.0	50.0	25.0	62.5	31.3	15.6
H (%)	0.0	100.0	100.0	75.0	87.5	87.5

^1^ HO = Holstein; F_1_ = first-generation crosses, i.e., HO dam × breed A sire or breed A dam × HO sire; G_2_ = second-generation crosses, i.e., F_1_ dam × breed B sire; G_3_ = third-generation crosses; G_4_ = fourth-generation crosses; G_5_ = fifth-generation crosses.

**Table 3 animals-11-03414-t003:** Description of the three datasets used for the statistical analyses of milk production (dataset 1), udder health performance (dataset 2) and reproduction performance (dataset 3). For each, the number of lactations and the number of cows for Holstein (HO) purebred, first and second generation of crosses (F_1_ and G_2_, respectively) obtained from three-breed rotational crossbreeding into purebred HO herds are indicated.

Genetic Class ^1^		Dataset 1 (Production Data)		Dataset 2 (Udder Health Data)		Dataset 3 (Fertility Data)	
		No. of Lactations	No. of Cows	No. of Lactations	No. of Cows	No. of Lactations	No. of Cows
HO	Primiparous	1226	1226	805	805	613	613
	Multiparous	2967	1377	1809	1048	1010	625
	Both	4193	1770	2614	1396	1623	945
F_1_	Primiparous	540	540	315	315	246	246
	Multiparous	1229	478	644	345	376	220
	Both	1769	606	959	461	622	338
G_2_	Primiparous	341	341	164	164	105	105
	Multiparous	459	225	210	131	99	66
	Both	800	354	374	218	204	128
Total	Primiparous	2107	2107	1284	1283	964	964
	Multiparous	4655	2080	2663	1524	1485	911
	Both	6672	2730	3947	2074	2449	1411

^1^ HO = purebred Holstein; F_1_ = first-generation crosses (50% HO); G_2_ = second-generation crosses (25% HO).

**Table 4 animals-11-03414-t004:** Herd compositions (combinations of the three genetic classes HO, F_1_ and G_2_) simulated to estimate the economic performance at herd level.

Herd Simulated	Percentage of Cows of Each Genetic Class in the Herd		
	HO	F_1_	G_2_
100HO	100	0	0
50HO-50F_1_	50	50	0
32HO-63F_1_-5G_2_	32	63	5
18HO-67F_1_-15G_2_	18	67	15
10HO-60F_1_-30G_2_	10	60	30

**Table 5 animals-11-03414-t005:** Least square means estimates (standard error in parentheses) of 305-d milk, fat and protein yields, fat and protein contents, somatic cell score (SCS), calving to first service interval (CFS) and calving interval (CI) for primiparous cows of the three genetic classes: Holstein (HO), first-generation crossbreds with 50% of Holstein genes (F_1_) and three-breed crossbreds of second generation with 25% of Holstein genes (G_2_).

Performance	Primiparous		
	HO (100% HO)	F_1_ (50% HO)	G_2_ (25% HO)
Milk yield (kg/cow)	7690 ^a^ (172.9)	7655 ^a^ (184.6)	7050 ^b^ (196.1)
Fat yield (kg/cow)	296 ^a^ (6.8)	310 ^b^ (7.3)	294 ^a^ (7.7)
Protein yield (kg/cow)	240 ^ab^ (5.5)	245 ^a^ (5.9)	232 ^b^ (6.3)
Fat content (g/kg/cow)	39.0 ^a^ (0.53)	41.2 ^b^ (0.57)	42.6 ^c^ (0.61)
Protein content (g/kg/cow)	31.3 ^a^ (0.31)	32.1 ^b^ (0.33)	33.1 ^c^ (0.35)
SCS (I.S.)	2.67 ^a^ (0.180)	2.69 ^a^ (0.198)	2.89 ^a^ (0.213)
CFS (days)	90 ^a^ (1.0)	78 ^b^ (1.0)	73 ^b^ (1.0)
CI (days)	422 ^a^ (1.0)	399 ^b^ (1.0)	397 ^b^ (1.0)

^a–c^ Means within a row with different superscript letters differ significantly (*p* < 0.05).

**Table 6 animals-11-03414-t006:** Least square means estimates (standard error in parentheses) of 305-d milk, fat and protein yields, fat and protein contents, somatic cell score (SCS), calving to first service interval (CFS) and calving interval (CI) for multiparous cows of the three genetic classes: Holstein (HO), first-generation crossbreds with 50% of Holstein genes (F_1_) and three-breed second-generation crossbreds with 25% of Holstein genes (G_2_).

Performance	Multiparous		
	HO (100% HO)	F_1_ (50% HO)	G_2_ (25% HO)
Milk yield (kg/cow)	7790 ^a^ (43.1)	7596 ^b^ (54.4)	6953 ^c^ (84.6)
Fat yield (kg/cow)	293 ^a^ (1.7)	301 ^b^ (2.1)	281 ^c^ (3.3)
Protein yield (kg/cow)	242 ^a^ (1.4)	243 ^a^ (1.7)	226 ^b^ (2.7)
Fat content (g/kg/cow)	38.0 ^a^ (0.13)	39.9 ^b^ (0.17)	40.7 ^c^ (0.26)
Protein content (g/kg/cow)	31.2 ^a^ (0.08)	32.1 ^b^ (0.10)	32.5 ^c^ (0.15)
SCS (I.S.)	2.88 ^a^ (0.051)	2.62 ^b^ (0.066)	2.83 ^ab^ (0.109)
CFS (days)	85 ^a^ (1.0)	73 ^b^ (1.0)	72 ^b^ (1.0)
CI (days)	402 ^a^ (1.0)	384 ^b^ (1.0)	389 ^ab^ (1.0)

^a–c^ Means within a row with different superscript letters differ significantly (*p* < 0.05).

## Data Availability

Data sharing not applicable.

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
