# Peer review of "Milk, Fertility and Udder Health Performance of Purebred Holstein and Three-Breed Rotational Crossbred Cows within French Farms: Insights on the Benefits of Functional Diversity"

_animals, 2021, doi:10.3390/ani11123414_

Round 1
Reviewer 1 Report
Overall:
Your paper addresses an interesting topic for crossbreeding in dairy cattle.
Minor comments:
L293: In dairy industry, increasing coefficient of inbreeding is one of the important issues.
Cross breeding is one of the methods to avoid inbreeding.
However, decreasing in the number of purebred population bring decreasing in genetic diversity in gene pool in purebred cattle.
How do you think about to keep genetic diversity of purebred populations?
L129: For SCC, did you use mean values in each lactation period.
L167: The effects of age (or age group) at calving affects lactation yields.
The mathematical model for genetic evaluation in your country also contains the age effects.
Why didn't you include it in your mathematical model(2) ?
Or did you use mature equivalent MYs?
L185: Back to original scale?
Author Response
Dear Reviewer 1,
Thank you for your comments.
Please find attached our responses.
Best regards,
Julien Quénon and Marie-Angélina Magne

Reviewer 2 Report
Overall I read with interest on your study with crossbreeding.
There are 2 major concerns that I have that should be addressed in the Discussion.
The study is based on HO, versus, F1 versus G2. However, in the discussion you do not frame your results on breeds involved in crossbreeding, breed complimentarity, or breed specific combining ability. I agree that this is not the premise of the study, but your discussion should include that the results could change depending on what breeds farmers use for crossbreeding. I have some specific comments. See below.
Also, Your results are only based off of production and some fertility. A crossbreeding program involves so much more than these. Please include some discussion on the results that your conclusions are only based on production and fertility
Line 14 and 37: What are the functional performances that you speak of?
Line 71: Should be purebred. One word
Line 72: what is "Holstein lead"
Line 102 and Figure 1: I cannot see the breed boxes in the graphs. They are too small to read. You might want to put these figures in the appendix so we can see them better, otherwise, they should be removed. They are too difficult to read.
Line 127: Where there crossbreds alongside Holsteins in all herds?
Line 131: How to get the 305 milk yield? How did you extrapolate? A linear regression? Woods curve? Random regression?
Line 139: What % was missing?
Line 157: Stat analysis. Did you have multiple observations per cow? Or was it 1 record per cow? If more than 1 record, I did not see cow as a random effect.
Line 277: I think you need some discussion on breeds used here. You had Simmental and Jersey is the crossbreeding rotation and they are very different and have very different results compared with Viking Red or Montbeliarde. How do breeds effect your results?
Line 341-352 I think this is Materials and methods because you are provided new ecnomoic analysis
Figure 2 and line 356-366: This should be in the Discussion
Line 398-401: This statement makes no sense. Farmers may experience loss of milk yield depending on the breeds used. Also, does it matter if they lose milk yield? What do French farmers get paid for? Probably not milk volume. I don't like the statement that says farmers shoul keep some Holsteins in the herd just because milk volume is lower. I think your results do not show this.
Figure A1: You might needs a footnote of abbreviations for the breeds listed.
Author Response
Dear Reviewer 2,
Thank you for your comments.
Please find attached our responses.
Best regards,
Julien Quénon and Marie-Angélina Magne

Round 2
Reviewer 2 Report
Authors have satisfactorily responded to my comments.